# Dietary Constituents: Relationship with Breast Cancer Prognostic (MCC-SPAIN Follow-Up)

**DOI:** 10.3390/ijerph18010084

**Published:** 2020-12-24

**Authors:** Trinidad Dierssen-Sotos, Inés Gómez-Acebo, Nuria Gutiérrez-Ruiz, Nuria Aragonés, Pilar Amiano, Antonio José Molina de la Torre, Marcela Guevara, Jessica Alonso-Molero, Mireia Obon-Santacana, Guillermo Fernández-Tardón, Ana Molina-Barceló, Juan Alguacil, Rafael Marcos-Gragera, Paz Rodríguez-Cundín, Gemma Castaño-Vinyals, Rosario Canseco Fernandez, Jesús Castilla, Amaia Molinuevo, Beatriz Pérez-Gómez, Manolis Kogevinas, Marina Pollán, Javier Llorca

**Affiliations:** 1Consortium for Biomedical Research in Epidemiology and Public Health (CIBER de Epidemiología Y Salud Pública-CIBERESP), 28029 Madrid, Spain; gomezai@unican.es (I.G.-A.); nuria.aragones@salud.madrid.org (N.A.); epicss-san@euskadi.eus (P.A.); ajmolt@unileon.es (A.J.M.d.l.T.); mp.guevara.eslava@navarra.es (M.G.); alonsomoleroj@gmail.com (J.A.-M.); mobon@idibell.cat (M.O.-S.); gfernanta@gmail.com (G.F.-T.); alguacil@dbasp.uhu.es (J.A.); rmarcos@iconcologia.net (R.M.-G.); gemma.castano@isglobal.org (G.C.-V.); jcastilc@navarra.es (J.C.); au-molinuevo@euskadi.eus (A.M.); bperez@isciii.es (B.P.-G.); manolis.kogevinas@isglobal.org (M.K.); mpollan@isciii.es (M.P.); llorcaj@unican.es (J.L.); 2IDIVAL Santander, 39011 Santander, Spain; m129@humv.es; 3Faculty of Medicine, University of Cantabria, 39011 Santander, Spain; arrancaespinos@hotmail.com; 4Epidemiology Section, Public Health Division, Department of Health of Madrid, 28009 Madrid, Spain; 5Public Health Division of Gipuzkoa, Health Department, BioDonostia Research Institute, 20014 San Sebastian, Spain; 6Grupo de Investigación en Interacciones Gen-Ambiente y Salud (GIIGAS), Instituto de Biomedicina (IBIOMED), Universidad de León, 24071 León, Spain; 7Navarra Public Health Institute, 31003 Pamplona, Spain; 8IdiSNA, Navarra Institute for Health Research, 31008 Pamplona, Spain; 9Unit of Biomarkers and Susceptibility, Oncology Data Analytics Program, Catalan Institute of Oncology (ICO), Hospitalet de Llobregat, 08908 Barcelona, Spain; 10Colorectal Cancer Group, ONCOBELL Program, Bellvitge Biomedical Research Institute (IDIBELL), Hospitalet de Llobregat, 08908 Barcelona, Spain; 11Health Research Institute of the Principality of Asturias, Oncology Institute, University of Oviedo, 33006 Oviedo, Spain; 12Cancer and Public Health Area, FISABIO—Public Health, 46020 Valencia, Spain; molina_anabar@gva.es; 13Centro de Investigación en Recursos Naturales, Salud y Medio Ambiente (RENSMA), Universidad de Huelva, 21007 Huelva, Spain; 14Research Group on Statistics, Econometrics and Health (GRECS), Universitat de Girona, 17071 Girona, Spain; 15Epidemiology Unit and Girona Cancer Registry, Oncology Coordination Plan, Department of Health, Autonomous Government of Catalonia, Catalan Institute of Oncology, 17007 Girona, Spain; 16Preventive Medicine, Hospital Universitario Marqués de Valdecilla, 39011 Santander, Spain; 17ISGlobal, 08036 Barcelona, Spain; 18Department of Public Health, Universitat Pompeu Fabra (UPF), 08002 Barcelona, Spain; 19IMIM (Hospital Del Mar Medical Research Institute), 08003 Barcelona, Spain; 20Servicio de Cirugía General y del Aparato Digestivo, Complejo Asistencial Universitario de León, 24001 León, Spain; rcanseco@saludcastillayleon.es; 21Cancer and Environmental Epidemiology Unit, National Centre for Epidemiology, Carlos III Institute of Health, 28029 Madrid, Spain

**Keywords:** breast cancer, dietary nutrients, overall survival, prognosis, mortality

## Abstract

The aim of this study was to characterize the relationship between the intake of the major nutrients and prognosis in breast cancer. A cohort based on 1350 women with invasive (stage I-IV) breast cancer (BC) was followed up. Information about their dietary habits before diagnosis was collected using a semi-quantitative Food Frequency Questionnaire. Participants without FFQ or with implausible energy intake were excluded. The total amount consumed of each nutrient (Kcal/day) was divided into tertiles, considering as “high intakes” those above third tertile. The main effect studied was overall survival. Cox regression was used to assess the association between death and nutrient intake. During a median follow-up of 6.5 years, 171 deaths were observed. None of the nutrients analysed was associated with mortality in the whole sample. However, in normal-weight women (BMI 18.5–25 kg/m^2^) a high intake of carbohydrates (≥809 Kcal/day), specifically monosaccharides (≥468 Kcal/day), worsened prognostic compared to lowest (≤352 Kcal/day). Hazard Ratios (HRs) for increasing tertiles of intake were HR:2.22 95% CI (1.04 to 4.72) and HR:2.59 95% CI (1.04 to 6.48), respectively (*p* trend = 0.04)). Conversely, high intakes of polyunsaturated fats (≥135 Kcal/day) improved global survival (HR: 0.39 95% CI (0.15 to 1.02) *p*-trend = 0.05) compared to the lowest (≤92.8 kcal/day). In addition, a protective effect was found substituting 100 kcal of carbohydrates with 100 kcal of fats in normal-weight women (HR: 0.76 95% CI (0.59 to 0.98)). Likewise, in premenopausal women a high intake of fats (≥811 Kcal/day) showed a protective effect (HR:0.20 95% CI (0.04 to 0.98) *p* trend = 0.06). Finally, in Estrogen Receptors (ER) negative tumors, we found a protective effect of high intake of animal proteins (≥238 Kcal/day, HR: 0.24 95% CI (0.06 to 0.98). According to our results, menopausal status, BMI and ER status could play a role in the relationship between diet and BC survival and must be taken into account when studying the influence of different nutrients.

## 1. Introduction

Breast cancer (BC) is the most frequently diagnosed cancer in women and the leading cause of cancer death worldwide [1,2] as well as in all European Countries [2]. In Europe, the overall survival rate of BC has improved in the last few decades. This favourable trend can be explained in part by both the advances in the treatment and management of BC over the last three decades [3,4], and, in women aged 50 and above, the development of screening programs. However, there are still noticeable differences between countries across Europe. Countries in Southern Europe (Spain and Portugal) and Nordic countries (Norway and Finland) show the lowest mortality rates, whereas countries in the Balkan Peninsula and parts of Eastern Europe (Hungary and Moldova) roughly double this mortality rate [5]. These differences have led to speculation that, beyond the major prognosis factors of BC, individual’s lifestyle (diet, physical exercise, etc.) may also play a key role in cancer survivorship [6].

Focusing on diet, several studies have proposed different mechanisms to explain the potential role of specific nutrients on cancer prognosis. Regarding fat intake, monounsaturated fats can have a protective effect due to their influence on tumor associated macrophages, promoting a change in polarization from M2 to M1 macrophages, thus leading to better prognosis [7]. Likewise, the consumption of foods rich in omega 3 such as bluefish, salmon, vegetable oils and nuts has been previously associated with better prognosis in women with node-negative disease [8]. Moreover, a high omega 3/omega 6 ratio has been associated with a better prognosis of BC [9], while saturated fats could have a deleterious effect on BC survival, by increasing circulating levels of endogenous estrogen, insulin-like growth factor (IGF)-1, and pro-inflammatory cytokines [10].

As for other nutrients, high animal-protein diets may accelerate insulin-like growth factor-1 (IGF-1) secretion, promoting cancer incidence and progression [11], and carbohydrate-rich diets could be associated with BC prognosis through different mechanisms. Monosaccharides could worsen BC prognosis by activating the insulin-IGF-1 axis and employing aerobic glycolysis as the primary energy harvesting pathway (Warburg effect). On the contrary, polysaccharides could improve BC prognosis by disrupting the insulin/IGF-1 axis, decreasing bioavailable androgenic and estrogenic factors, increasing fecal excretion of carcinogens, and modulating the gastrointestinal microbiota [12].

Improving the knowledge of the role of diet on BC prognosis will allow women to take part in increasing their chances of survival by adopting healthier habits. However, up to this date, the research on the influence of specific nutrients on BC prognosis has not obtained consistent findings [10]. The last report of the Continuous Update Project suggests that a diet rich in fiber (both before and after diagnosis), and soy (after diagnosis) could be associated with better BC survival. In contrast, a high intake of fats before developing the disease, especially saturated fats, seems to be associated with poorer survival rate [13]. Due to the lack of consistency in the results, additional research in this topic is required. Besides, the influence of potential interaction factors such as BMI or hormonal receptors has been scarcely studied to date. Menopausal status has been identified as an effect modifier in the relationship between fiber intake and BC prognosis [13] and between carbohydrates [14] and BC risk. Regarding hormone receptors, some studies have suggested that lifestyle risk factors, including diet, may have a comparatively greater influence in the development of ER− subtype than ER+ [15]. The Women’s Intervention Nutrition Study (WINS) [16], found a positive effect of reducing dietary fat intake on relapse-free survival of BC, and this effect was mainly noted in the subgroup with ER− tumors. Two large studies had identified obesity as an effect modifier in the relation between diet and BC: the E3N French study [14] aimed to investigate the effect of carbohydrates on BC risk, and the Women’s Health Initiative Randomized Controlled Trial focus aimed to do the same on fats [17]. Thus, it has been speculated that the relation between nutritional factors and BC could be modulated by obesity because of the effect of adiposity and, in particular, central adiposity on insulin resistance [14]. In this context, the aim of our study is to characterize the relationship between the intake of the major nutrients and prognosis in BC related to each women’s specific characteristics.

## 2. Materials and Methods

### 2.1. Study Population

The MCC-Spain breast cancer follow-up study is a prospective cohort study that comes from the MCC-Spain project. This project began in 2008 with the aim of investigating the relationship between genetic and environmental exposures and cancer development [18]. In 2016, the MCC-Spain project turned towards the identification of factors associated with cancer prognosis using the incident cases originally recruited between 2008 and 2013. Our initial cohort consists of 1685 incident BC cases recruited, and follow-up in 18 hospitals of 10 Spanish provinces (Asturias, Barcelona, Cantabria, Girona, Gipuzkoa, Huelva, León, Madrid, Navarra and Valencia) in the MCC-Spain project. Further information can be found elsewhere [19,20]. From this initial cohort we exclude the following participants: (1) those with non-invasive tumors, (2) those without food frequency questionnaire (FFQ) and (3) those with implausible energy intakes (<750 or ≥4500 kcal) in the FFQ. 1350 women fulfilled the inclusion criteria of the current analysis. The process is summarized in the study flow-chart (Figure 1).

### 2.2. Ethical Approval

The Ethics Committees of participating hospitals approved the study protocols of MCC-Spain [18]. (code for the ethics committee of Cantabria: September 2016) and participants were provided with written informed consent at the time of their enrolment in the study, which also included the authorization for following up the patients via medical records or phone calls; only participants agreeing to being followed up were included in the inception cohorts. The database was registered with the Spanish Agency for Data Protection, number 2102672171. Permission to use the study database will be granted to researchers outside the study group after revision and approval of each request by the Steering Committee.

### 2.3. Dietary and Prognosis Factors Assessment

The information on pre-diagnosis diet was collected using a food-frequency questionnaire (FFQ) previously validated for the Spanish population [21]. In the FFQ, patients were asked about their eating habits during the previous year. The FFQ included 140 items. Each participant received the FFQ at the end of the interview in paper format, to be completed at home and returned by mail. The food composition table was a compiled table from CESNID Food Composition Table [22]. Cross-check questions were used to adjust the frequency of foods eaten and reduce overreporting of food groups with large numbers of items [23]. The food composition table was used to obtain daily intake of main nutrient groups: carbohydrates, proteins and fats. Besides, carbohydrates and proteins were subdivided into two subtypes according to their type (monosaccharide or polysaccharide) or their source (animal or vegetable) respectively, and fats into three (monounsaturated, saturated and polyunsaturated). The total amount consumed of each nutrient (Kcal/day) was divided into tertiles.

Covariate assessment information which may specifically be related to BC prognosis includes: sociodemographic factors (age at diagnosis, education attained, socioeconomic status), family history of breast cancer, anthropometric data and lifestyle (tobacco, physical activity: metabolic equivalents, constructed with data on reported level of physical activity during the last five years—excluding the one year before the interview). This information was gathered in a face-to-face interview using a standardized questionnaire administered by trained personnel. The clinical and anatomopathological characteristics of the tumor and the systemic treatment received were obtained by review of the medical records. The clinical-pathological data form includes tumor location and size, invasiveness, histological type, degree of differentiation, vascular or lymphatic infiltration, number of affected lymph nodes, infiltration of resection margins, hormonal receptors, C-ErbB2 by immunohistochemistry or Fluorescent in situ hybridization ( FISH), Ki-67, clinical TNM (before treatment) and surgical stage. The Pathological Prognosis Stage (PPS) was calculated in those women with BC treated with surgery as initial treatment; this includes clinical staging information, findings at surgery and pathological findings [24]. Finally, systemic treatment received includes chemotherapy, hormone therapy, and immunotherapy.

### 2.4. Follow-Up and Ascertainment of Events

Follow-up was carried-out between 2017 and 2018 by reviewing medical records and contacting the participants by telephone. The main effect was death. Events of death were consulted in the medical record or in the Spanish National Death Index (a nation-wide data-base supported by the Spanish Ministry of Health) when medical records do not provide them, and in women whose last contact with the hospital had occurred three or more months before our revision of her medical record [25].

### 2.5. Statistical Analysis

An overall survival analysis was carried out in which the main effect variable was all causes of death, and patients who remain alive after the end of follow-up were considered censored. Withdrawals were censored at the date of last contact with the patient. Time of follow-up was calculated as the difference between date of diagnosis and date of death or date of last contact. Cox regression was used to assess the association between death and nutrient intake; results are reported as hazard ratio (HR) with confidence interval (95% CI). Three different types of adjustment were made; for age, total energy intake (Kcal/day) [26], and the main clinical factors related to BC prognosis, hospital of recruitment, PPS and the systemic treatment received by the patients (chemotherapy, hormone therapy or immunotherapy). PPS could only be established for women without neoadjuvant therapy. Therefore, when adjusting for PPS two additional categories were included: one for women with neoadjuvant therapy and another for those women whose PPS could not be determined, in spite of having no neoadjuvant therapy. In the second model, anthropometric data (BMI), lifestyle factors (consumption of tobacco and metabolic equivalents—METS) and sociodemographic factors (socioeconomic score and education) were added to the variables adjusted for in the first model; and in the third model, a specific adjustment by nutrients was added to the previous ones. Each type of main group of nutrients (carbohydrates, proteins and fats) was adjusted for the percentage calories from the other types. Finally, mutual adjustment was subsequently added for animal protein and plant protein, short and long chain carbohydrate, and the different type of fats, respectively. This strategy of adjustment simulates an energy substitution model. Lastly, a stratified analysis was carried out by three potential effect modifiers: menopausal status at diagnosis, body mass index (BMI) and Estrogen receptors (ER). Two-tailed P values for linear trend tests across tertiles were calculated.

In order to find out the consequences of replacing nutrients on BC survival, a substitution analysis was carried out. The effect of substituting one type of nutrient for another was calculated by estimating a substitution model with constant total energy consumption, following the method proposed by Willet et al. [26,27].

To do so, the initial model was:$$log\;=\;+\beta_{1}\times carbohydrates+\beta_{2}\times proteins+\beta_{3}\times fats+\beta_{4}\times total\;energy+\beta_{5}\times covariates$$

This model cannot be estimated because of the linear relationship between the regressors (i.e., total nutrients = carbohydrates + proteins + fats).

By omitting a term, say carbohydrates, the estimated model indicates the effect of substituting an amount of carbohydrates by equal amounts of proteins or fats, as the model keeps the total energy intake constant. This method is an iso-temporal model that allows the estimation of the effect of substituting a fixed amount of a type of nutrient for another.

Our sample size (1350 women) had 85% power to detect hazard ratio >1.7 or <0.59 (=1/1.7).

We have carried out a sensitivity analysis excluding advanced BC (stage IV) at diagnosis. Analyses were performed using the package Stata 16/SE (Stata Corp, College Station, TX, USA).

## 3. Results

A total of 1350 women diagnosed with BC were included in this study. The median follow-up time was 6.5 years. 171 deaths occurred during follow-up. These women had, at enrolment, an average age of 55.84 years (SD: 12.5); 487 cases (36.1%) were premenopausal women. Appendix A summarizes the main characteristic of the tumors. 63.2% tumors were luminal A-like, and 19.3% luminal B-like. The majority of cases were diagnosed at early stages. Fifty-two of the 726 women diagnosed in early stages (I–II) had BC recurrence. Appendix A summarizes the dietary constituents in the whole cohort and in the two subgroups of women who died and survived at the end of follow-up. The mean intake of carbohydrates, proteins, and fats were respectively (192.6 g/day, 80.2 g/day, and 83.2 g/day). No significant differences were found in the intake of nutrients between women who died and those who survived. Appendix A shows the main characteristics of the study population according to tertiles of energy intake. In those women with high energy intake (third tertile), there was a higher percentage of premenopausal women. In women with low energy intake, lower percentages of university studies and a high socioeconomic status was found. No differences in previous use of hormonal contraceptive or hormone replacement therapy were found. On average, women with higher energy intake were younger and had lower BMI and higher alcohol intake.

### 3.1. Relationship between Carbohydrate Intake and Overall Survival

Although carbohydrate intake or its different subtypes did not have an effect on survival in the whole sample with BC (Table 1), in normal-weight women (BMI ≤ 25) high consumption of carbohydrate showed a strong association with mortality irrespectively of the model of adjustment (HR: 3.36 95% CI (1.01 to 11.2) in the third model). This deleterious effect was restricted to monosaccharides intake showing a significant linear trend (HR: 2.22 95% CI (1.04 to 4.72) for moderate intake and HR: 2.59 95% CI (1.04 to 6.48) for high intake, *p* trend = 0.04). On the other hand, in overweight or obese women no relationship was found between total carbohydrates intake and mortality (HR: 1.19 95% CI (0.48 to 2.95) (Figure 2). However, moderate and high intake of monosaccharides showed protection in the first (HR:0.50 95% CI (0.29 to 0.89) for moderate intake and HR: 0.45 95% CI (0.23 to 0.91) for high intake, *p* trend = 0.03); and the second model of adjustment (HR: 0.53 95% CI (0.29 to 0.95) for moderate intake and HR: 0.50 95% CI (0.24 to 1.02) for high intake, *p* trend = 0.06), but this finding did not reach statistical significance, adjusting the effect for polysaccharides intake and diabetes (model 3 HR: 0.60 95% CI (0.33 to 1.10) for moderate intake, and HR: 0.57 95% CI (0.22 to 1.19) for high intake, *p* trend = 0.14) (Figure 2). In contrast, menopausal status and ER did not modify the effect observed in the whole population (Figure 2, Appendix A).

### 3.2. Relationship between Protein Intake and Overall Survival

Total protein intake or its subtypes were not associated with the overall survival of BC (Table 1). However, when we analyze animal protein intake separately, we observed a differential effect according to menopausal status. In postmenopausal women, we found a higher risk associated with moderate intake (HR: 1.88 95% CI (1.17 to 3.04)) that disappeared in premenopausal women. In contrast, stratifying by ER, in ER-tumors a high animal protein intake showed an inverse association with BC mortality (HR: 0.24 95% CI (0.06 to 0.99)). No interaction with BMI was observed (Figure 3). On the other hand, a moderate intake of vegetable protein diminished the risk of dying among premenopausal women regardless of the model of adjustment used (HR:0.35 95% CI (0.14 to 0.86) in the third model), which disappeared in postmenopausal women. No differential effect was observed according to BMI or ER status (Figure 3, Appendix A).

### 3.3. Relationship between Fat Intake and Overall Survival

No relationship was found between total fat intake or its different subtypes and BC survival, irrespective of the model of adjustment used (Table 1). However, analyzing separately by menopausal status (Figure 4), a moderate intake of fat shows a protective effect confined to premenopausal women and only in the third model of adjustment ((HR: 0.25 95% CI (0.08 to 0.84) for moderate intake and HR: 0.20 95% CI (0.04 to 0.98) for high intake, *p* trend = 0.06). On the other hand, in normal-weight women, polyunsaturated fat showed a protective effect with a linear trend regardless the model of adjustment (HR: 0.39 95% CI (0.15 to 1.02), *p* trend = 0.05 in the third model) (Figure 4). No differential effect was observed according to the ER status (Figure 4, Appendix A).

No relevant differences were found in the sensitivity analysis (excluding stage IV tumors), therefore these results are not shown.

### 3.4. Effect of Substituting a Nutrient for Another

Finally, we investigated the effect of substituting 100 kcal of a type of nutrient with 100 kcal of another type (Table 2). No association was observed in the whole population; however, a protective effect was found when substituting 100 kcal of carbohydrates with 100 kcal of fats in normal weighted women (HR: 0.76 95% CI (0.59 to 0.98) (Table 3).

For instance, taking 100 kcal more as proteins instead of 100 kcal as carbohydrates gives HR = 0.94, 95% CI 0.67–1.33. Adjusted for: hospital of recruitment, age, PPS score (0, IA, IB, IIA, IIB, IIA, IIB, IIC, IV, missing), systemic treatment received by the patients: chemotherapy (yes, non), hormone therapy (yes, non) and immunology therapy (yes, non), total energy intake (Kcal/day) one year before the diagnosis, menopausal status, socioeconomic status (low, middle, high), education attained, physical activity (metabolic equivalents (METs)) during the 5 years before diagnosis, smoking status one year before recruitment (never; former; current) and Body Mass Index (kg/m^2^).

## 4. Discussion

Observational data has consistently linked poor dietary quality [28] and other lifestyle factors [6,29] to increased risk of developing and dying from malignancy. However, even though our study does not directly address the effects of lifestyle, our results do not reveal any direct effect on BC survival of the different types of nutrients evaluated. Interestingly, menopausal status, BMI and ER status could interact to modulate this influence.

We observed that a diet high in carbohydrate, in normal-weight women (BMI ≤ 25), and moderate in animal-protein in postmenopausal women, worsens BC survival. In contrast, some types of nutrients improve survival in specific subsets of women. In premenopausal women, a moderate intake of vegetable proteins and total fats seems to have a protective effect. On the other hand, in normal-weight women, polyunsaturated fats show protection with a significant trend.

With carbohydrates, although different mechanisms through which they could impact on cancer progression have been identified [30], most of the epidemiological studies developed to date [31,32,33] and the International World Cancer Research Fund meta-analysis [34] have obtained inconclusive results. In the same line, we did not obtain an association between a diet high in carbohydrates and BC prognosis. However, in normal-weight women our data show an increased risk associated with carbohydrate intake, and more specifically to monosaccharide intake. Besides, when substituting 100 g of carbohydrates for 100 g of fats, a protective effect was observed limited to this subgroup of women. A possible explanation of the deleterious effect of a high carbohydrate diet could be the increase of circulating insulin and consequently associated IGF-1, which has been related to both the initiation and the progression of BC [35]. In addition, simple carbohydrates may increase risk of cancer by employing aerobic glycolysis as the primary energy harvesting pathway [12]. Nevertheless, it is somewhat paradoxical that, according to our results, the effect of carbohydrate intake is observed only among those women with normal weight, since a plausible mechanism in the relationship between carbohydrate intake and cancer survival is the impaired intolerance to glucose frequently associated with obesity [36]. Besides, the serum concentrations of inflammatory cytokines, glucose, insulin and free IGF-1 are notably elevated in obese patients and provide a pro-tumorigenic environment that may explain the worse prognosis of obese patients with cancer. In relation to the lack of influence of the menopausal status and ER tumors on this relationship, to the best of our knowledge no other studies have been published taking into account menopausal or ER status in the relationship between carbohydrates and BC prognosis. However, ER status has been identified as an effect modifier in the relationship between IGF-1 levels and BC risk, restricting the effect of IGF-1 levels to those ER positive tumors [37].

According to our results, protein intake does not seem to have a prognostic effect in BC women. Consistent with this finding, several prospective studies evaluating the effect of protein intake before diagnosis did not obtain significant results [33,38,39,40,41]. However, some studies focusing on the role of a high protein diet after diagnosis found a survival advantage [32,42,43,44,45]. When separately analyzing the source (animal or vegetable) of proteins consumed, we identified interaction with menopausal and ER status. Among postmenopausal women, we observed an increase in the mortality rate associated with a moderate intake of protein from animal sources. Previous prospective studies aimed to evaluate mortality in different populations have also found higher mortality rates associated with animal protein intake [11,46,47,48]. It has been speculated that a high protein animal-based diet may accelerate IGF-1 secretion, which may promote cancer progression [11]. In contrast, the Nurses’ Health Study, a large prospective study including women with early stages of BC [42], showed a lower risk of death resulting from BC in those with higher animal protein intake after diagnosis, but this study did not make a stratified analysis according to menopausal status and thus its results are not directly comparable with ours. Regarding the influence of vegetable protein intake, we observe a protective effect in those with a moderate intake but restricted to premenopausal women. This protection could be related to an improved insulin sensitivity in diets rich in plant proteins [49]. Consistent with our results, Song et al. [47], found that high plant protein intake was inversely associated with all-cause mortality in a cohort of US health care professionals. However, other studies focusing on BC prognosis show contradictory results, McEligot et al. found that the intake of vegetables improves overall survival among postmenopausal women [33] while the Nurses’ Health Study [42] failed to find associations. Our data show a higher intake of vegetable proteins than that obtained by Holmes (the median value of the fifth quintile: 36.6 g/day vs. 25.0 g/day), which could explain the differences between both studies and the lack of association reported in the Nurses´ Health Study. On the other hand, according to our results, the ER status seems to modify the effect of animal protein intake. In those women with ER negative tumors a high intake of animal protein was associated with a strong protection. This result must be interpreted cautiously given the small number of events in this subgroup. We are not aware of any other study evaluating this interaction; however, Holmes et al. analyzing the association between protein intake and risk of distant recurrence did not found interactions by ER status [42].

Finally, regarding fat intake, two randomized trials addressed the impact of dietary modification (a low-fat dietary intervention) on BC recurrence and mortality, with mixed results. The Women’s Intervention Nutrition Study (WINS) [16], a large randomized trial based on postmenopausal women with an early-stage BC diagnosis, obtained a longer relapse-free survival in the intervention group (women who followed a diet with a less total fat intake). However these results were not confirmed in the Women’s Healthy Eating and Living Study (WHEL) [8]. In our research, no relationship with overall survival was found with either total fat intake or its different types in the whole cohort. In line with our results, several prospective studies focusing on both pre-diagnosis [38,50,51] and post-diagnosis diet [8,9,16,52,53] did not find association with fat intake. In contrast, the Women’s Health Initiative randomized clinical trial found that those postmenopausal women randomized to a low-fat dietary pattern had increased BC overall survival [17]. Considering the role of potential interactions, menopausal status and BMI seem to modify the effect of dietary fats intake. In premenopausal women, we found a protective effect of both a moderate and high intake of fats (*p* trend = 0.06). We are not aware of other studies focusing on premenopausal women, but it has been suggested that the association between dairy intake (rich in saturated fat) and BC risk may differ by menopausal status, showing an inverse association in premenopausal women [54]. On the other hand, in normal-weight women we found protection associated with polyunsaturated intake, with a significant trend. Polyunsaturated acids seem to have an opposite effect on carcinogenesis and tumoral progression according to their omega-3/omega-6 composition. Diets high in omega-3 appear to have anti-inflammatory effects, whereas omega-6-derived eicosanoids have proinflammatory effects [55].Consistently with this hypothesis, a protective effect has been previously found with omega-3 fatty acids [8]. Although we were not able to analyze the omega3/omega6 composition separately, Spanish women were previously identified as the largest consumers of fish (food rich in omega3) in the European Prospective Research on Cancer and Nutrition (EPIC). [56,57]. To the best of our knowledge this is the first study evaluating the prognostic effect of diet on BC developed in a Spanish population. Our results may in part be due to differences in the contributing foods of fats in different study population [58]. In this regard a review focused on the influence of dietary fats on health underlined that the types of food supplying the nutrients are more important than the total amount of fats [59].

Our study has several strengths that should be highlighted. First, we include a separate analysis according to potential effect modifiers (menopausal status, BMI and ER receptors), which only few other studies have previously considered. Secondly, the use of a FFQ (validated for a Spanish population) to assess diet information facilitates the comparison of our results with those obtained in other studies, given that it is still the primary dietary assessment tool in epidemiological studies [60]. Finally, the collection of diet information before the diagnosis of BC rules out the introduction of information biases, since the diet in the first year after diagnosis is strongly influenced by the treatment received [61]. Despite these strengths, there are some weaknesses in our study that should be mentioned. First, the use of FFQ could be affected by recall bias but, if it exists, it would probably be non-differential, underestimating the associations studied. Besides, the lack of longitudinal data on dietary consumption prevents the inclusion of dietary changes in the analysis. This is relevant in our BC cohort, given that several studies found changes in diet after a BC diagnosis [62,63]. Secondly, statistical power was limited for the subgroup analyses, so we were not able to evaluate the effect of nutrients intake according to tumor subtype. Finally, our study population was followed up for more than six years after diagnosis; however, BC survival is often considerably longer, thus a six-year follow-up only allowed us to investigate risk of death relatively close to diagnosis.

## 5. Conclusions

To sum up, according to our results, pre-diagnosis intake was not associated with a greater risk of dying in BC women. However, it may have a relevant role in some sub-groups of women. These findings, if confirmed, will allow BC survivors to have a more active role in their health care. This is especially relevant in premenopausal women for whom the advantages of screening programs remain uncertain. However, more prospective studies are needed to confirm our results.

## Figures and Tables

**Figure 1 ijerph-18-00084-f001:**
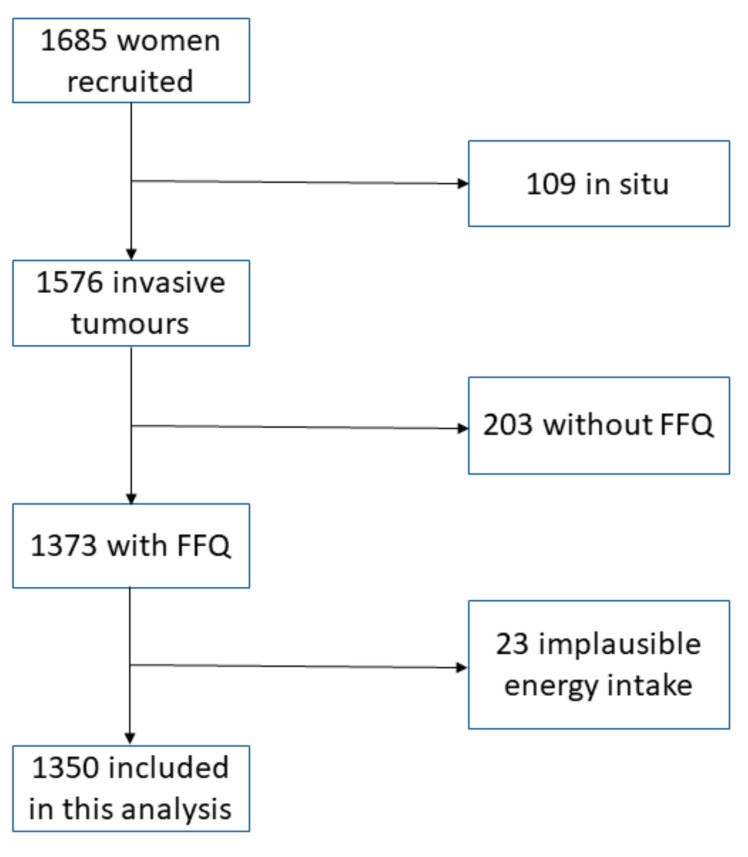
Study flowchart.

**Figure 2 ijerph-18-00084-f002:**
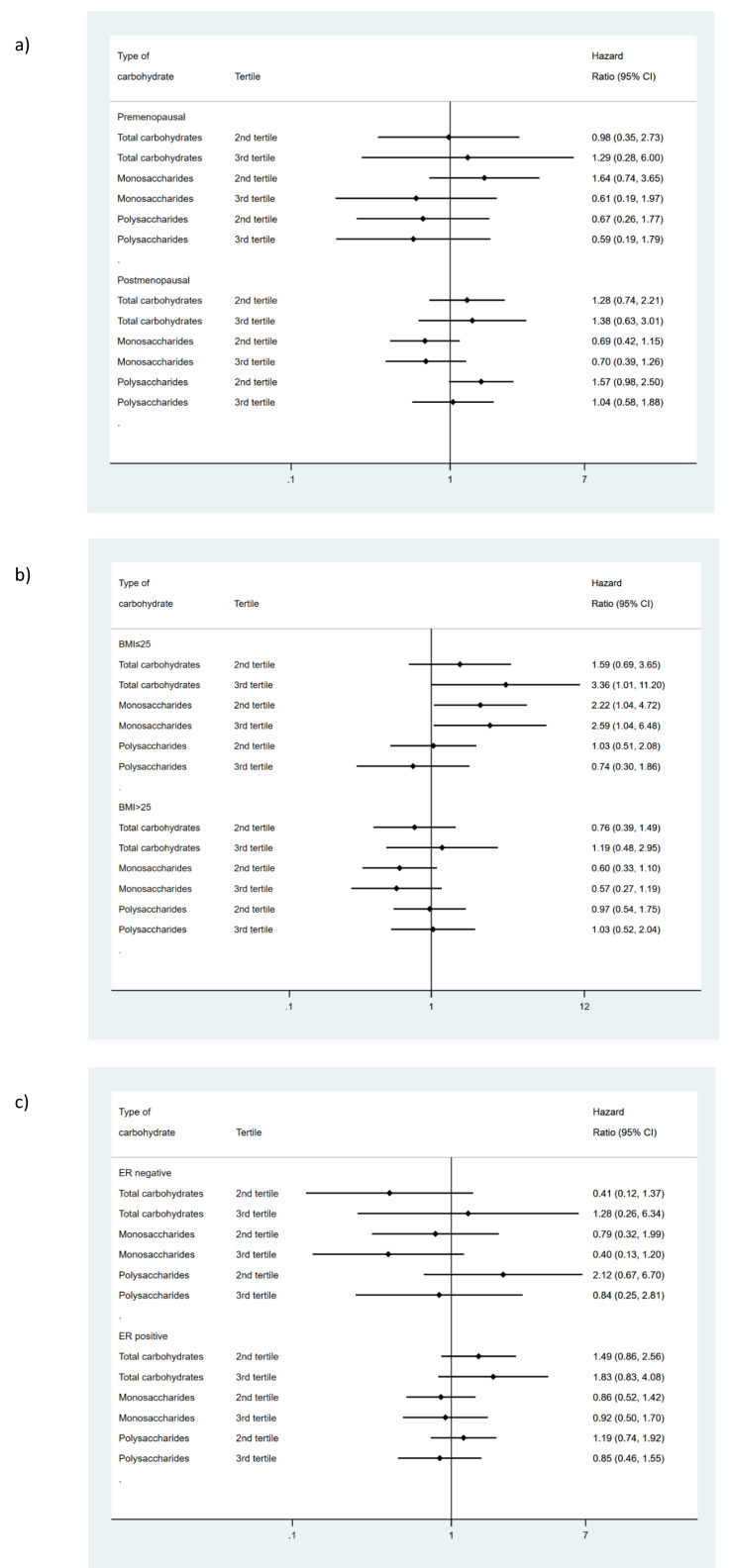
Influence of carbohydrate intake on breast cancer survival according to (**a**) menopausal status, (**b**) BMI and(**c**) estrogen receptor status. Hazard ratios comparing third and second tertiles vs. first tertile of consumption. Adjusted for hospital of recruitment, age, PPS score (0, IA, IB, IIA, IIB, IIA, IIB, IIC, IV, non-applicable, missing), systemic treatment received by the patients: chemotherapy (yes, non), hormone therapy (yes, non) and immunology therapy (yes, non) and total energy intake (Kcal/day) one year before the diagnosis, socioeconomic status (low, middle, high), education attained, physical activity (metabolic equivalents (METs)) during the 5 years before diagnosis, smoking status one year before recruitment (never; former; current) and Body Mass Index (kg/m^2^), diabetes (yes, non, unknown) and Total carbohydrates intake was adjusted for the percentage of calories from the other major nutrient groups; mono-saccharides and poly-sacharides were mutually adjusted for percentage of calories.

**Figure 3 ijerph-18-00084-f003:**
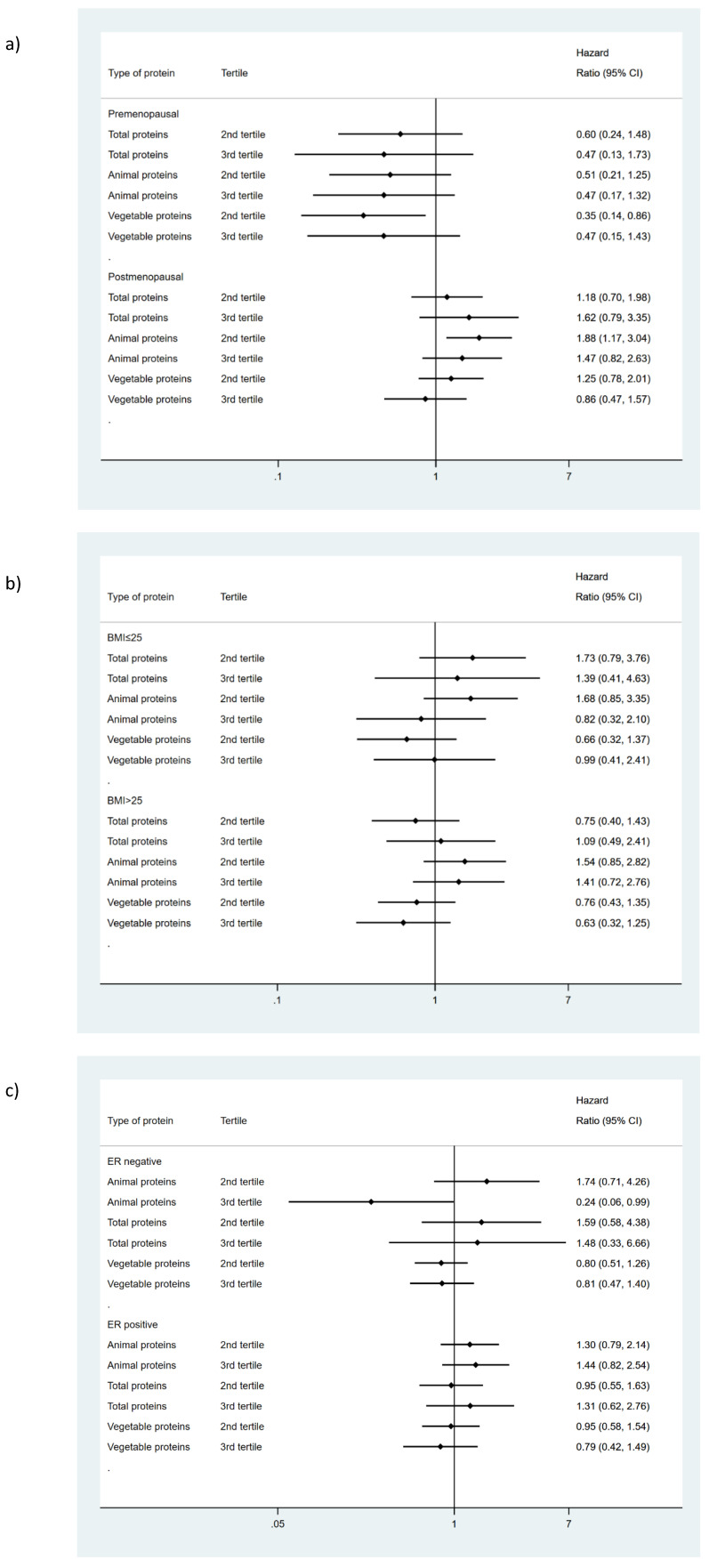
Influence of protein intake on breast cancer survival according to (**a**) menopausal status, (**b**) BMI and (**c**) estrogen receptor status. Hazard ratios comparing third and second tertiles vs. first tertile of consumption. Adjusted for hospital of recruitment, age, PPS score (0, IA, IB, IIA, IIB, IIA, IIB, IIC, IV, non-applicable, missing), systemic treatment received by the patients: chemotherapy (yes, non), hormone therapy (yes, non) and immunology therapy (yes, non), total energy intake (Kcal/day) one year before the diagnosis, socioeconomic status (low, middle, high), education attained, physical activity (metabolic equivalents (METs)) during the 5 years before diagnosis, smoking status one year before recruitment (never; former; current) and Body Mass Index (kg/m^2^), diabetes (yes, non, unknown). Total protein intake was adjusted for the percentage of calories from the other major nutrient groups. Animal and vegetable proteins were mutually adjusted for percentage of calories.

**Figure 4 ijerph-18-00084-f004:**
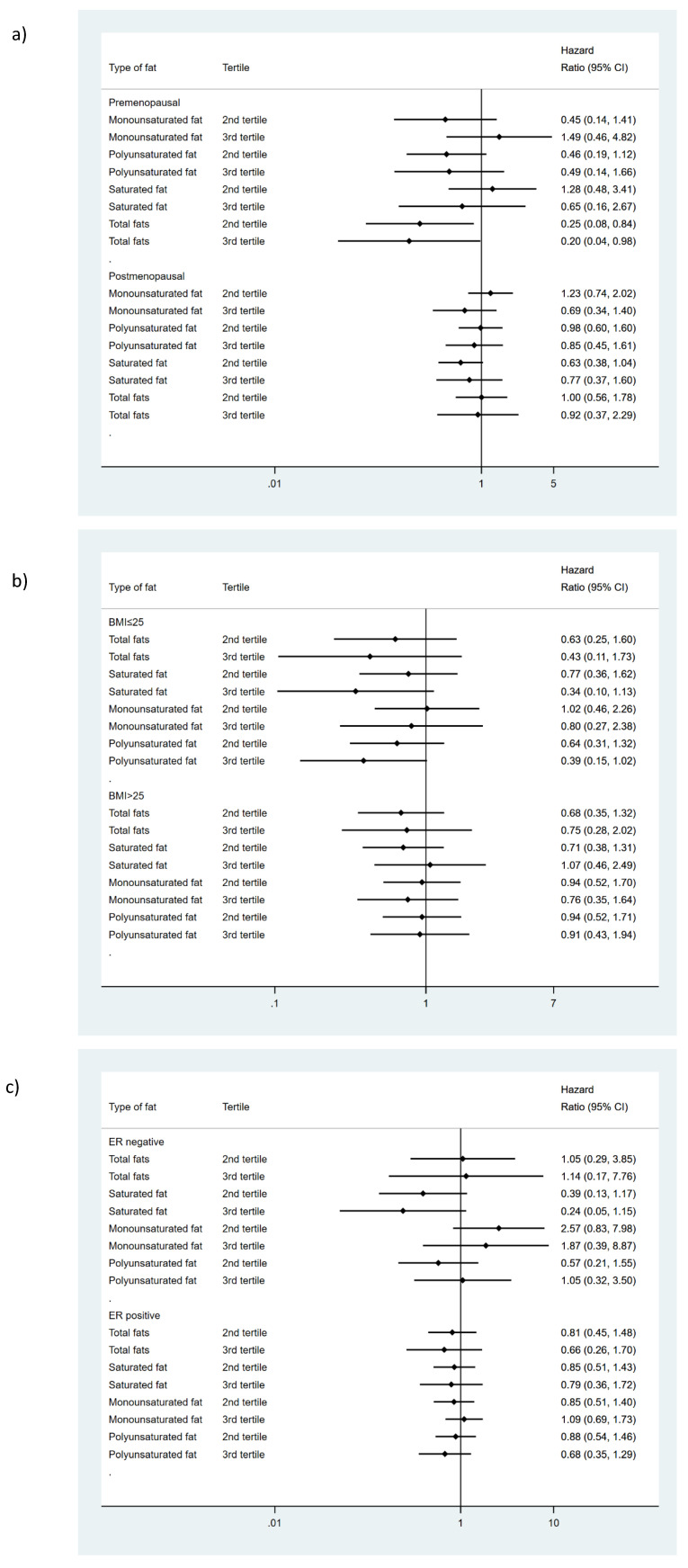
Influence of fat intake on breast cancer survival according to (**a**) menopausal status, (**b**) BMI and (**c**) estrogen receptor status. Hazard ratios comparing third and second tertiles vs. first tertile of consumption. Adjusted for hospital of recruitment, age, PPS score (0, IA, IB, IIA, IIB, IIA, IIB, IIC, IV, non-applicable, missing), systemic treatment received by the patients: chemotherapy (yes, non), hormone therapy (yes, non) and immunology therapy (yes, non) total energy intake (Kcal/day) one year before the diagnosis, socioeconomic status (low, middle, high), education attained, physical activity (metabolic equivalents (METs)) during the 5 years before diagnosis, smoking status one year before recruitment (never; former; current) and Body Mass Index (kg/m^2^), diabetes (yes, non, unknown). Total fat intake was adjusted for the percentage of calories from the other major nutrient groups. Saturated, monounsaturated and polyunsaturated fats were mutually adjusted for percentage of calories.

**Table 1 ijerph-18-00084-t001:** Influence of nutrient intake on breast cancer survival. Hazard ratios comparing third and second tertiles vs. first tertile of consumption.

Type of Nutrient		1st Tertile	HR (95% CI) 2nd Tertile	HR (95% CI) 3rd Tertile	*p* Trend
**Total carbohydrates**	Deaths/woman-years	52/2933	53/2946	66/2913	
	Model 1	1(ref.)	1.10 (0.72 to 1.68)	1.36 (0.80 to 2.33)	0.27
	Model 2	1(ref.)	1.09 (0.71 to 1.67)	1.31 (0.76 to 2.26)	0.34
	Model 3 *	1(ref.)	1.12 (0.71 to 1.75)	1.42 (0.75 to 2.69)	0.30
**Monosaccharides**	Deaths/woman-years	56/2910	55/2943	60/2939	
	Model 1	1(ref.)	0.86 (0.58 to 1.29)	0.76 (0.47 to 1.24)	0.28
	Model 2	1(ref.)	0.86 (0.58 to 1.30)	0.75 (0.45 to 1.22)	0.24
	Model 3 #	1(ref.)	0.89 (0.59 to 1.35)	0.74 (0.45 to 1.23)	0.25
**Polysaccharides**	Deaths/woman-years	54/2984	65/2870	52/2937	
	Model 1	1(ref.)	1.25 (0.85 to 1.85)	0.94 (0.58 to 1.52)	0.84
	Model 2	1(ref.)	1.23 (0.83 to 1.83)	0.96 (0.59 to 1.55)	0.91
	Model 3 $	1(ref.)	1.24 (0.83 to 1.84)	0.92 (0.56 to 1.50)	0.77
**Total proteins**	Deaths/woman-years	60/2888	52/2956	59/2948	
	Model 1 **	1(ref.)	0.89 (0.59 to 1.34)	0.96 (0.56 to 1.62)	0.84
	Model 2	1(ref.)	0.88 (0.58 to 1.33)	0.96 (0.56 to 1.64)	0.84
	Model 3	1(ref.)	0.94 (0.61 to 1.45)	1.12 (0.61 to 2.06)	0.77
**Animal proteins**	Deaths/woman-years	52/2928	64/2902	55/2962	
	Model 1	1(ref.)	1.37 (0.92 to 2.04)	1.13 (0.70 to 1.82)	0.60
	Model 2	1(ref.)	1.37 (0.91 to 2.06)	1.09 (0.68 to 1.76)	0.72
	Model 3 ¥	1(ref.)	1.35 (0.89 to 2.03)	1.05 (0.64 to 1.71)	0.84
**Vegetable proteins**	Deaths/woman-years	62/2894	52/2965	57/2932	
	Model 1	1(ref.)	0.88 (0.59 to 1.32)	0.75 (0.46 to 1.23)	0.26
	Model 2	1(ref.)	0.91 (0.61 to 1.37)	0.79 (0.48 to 1.30)	0.35
	Model 3 £	1(ref.)	0.90 (0.60 to 1.35)	0.77 (0.46 to 1.28)	0.32
**Total fats**	Deaths/woman-years	59/2902	59/2913	53/2976	
	Model 1	1(ref.)	0.77 (0.51 to 1.17)	0.70 (0.37 to 1.30)	0.22
	Model 2	1(ref.)	0.80 (0.52 to 1.22)	0.73 (0.39 to 1.37)	0.30
	Model 3 **	1(ref.)	0.73 (0.45 to 1.19)	0.63 (0.30 to 1.34)	0.22
**Saturated fats**	Deaths/woman-years	63/2909	56/2909	52/2974	
	Model 1	1(ref.)	0.79 (0.52 to 1.20)	0.76 (0.41 to 1.40)	0.33
	Model 2	1(ref.)	0.78 (0.51 to 1.18)	0.73 (0.39 to 1.36)	0.27
	Model 3 §	1(ref.)	0.74 (0.48 to 1.14)	0.70 (0.37 to 1.31)	0.22
**Monounsaturated fats**	Deaths/woman-years	56/2901	64/2926	51/2965	
	Model 1	1(ref.)	0.95 (0.63 to 1.43)	0.74 (0.43 to 1.28)	0.29
	Model 2	1(ref.)	0.99 (0.65 to 1.49)	0.77 (0.44 to 1.33)	0.36
	Model 3 ¶	1(ref.)	1.00 (0.66 to 1.53)	0.78 (0.44 to 1.38)	0.41
**Polyunsaturated fats**	Deaths/woman-years	60/2855	61/2984	50/2952	
	Model 1	1(ref.)	0.87 (0.58 to 1.30)	0.76 (0.45 to 1.28)	0.3
	Model 2	1(ref.)	0.90 (0.59 to 1.35)	0.81 (0.48 to 1.37)	0.44
	Model 3 ß	1(ref.)	0.86 (0.56 to 1.31)	0.77 (0.45 to 1.32)	0.34

HR: Hazard ratio. 95% CI: 95% confidence interval. Model 1: Adjusted for hospital of recruitment, age, PPS score (0, IA, IB, IIA, IIB, IIA, IIB, IIC, IV, non-applicable, missing), systemic treatment received by the patients: chemotherapy (yes, non), hormone therapy (yes, non) and immunology therapy (yes, non) and total energy intake (Kcal/day) one year before the diagnosis. Model 2: Adjusted for all previous variables and socioeconomic status (low, middle, high), education attained, physical activity (metabolic equivalents (METs)) during the 5 years before diagnosis, smoking status one year before recruitment (never; former; current) and Body Mass Index(kg/m^2^). Model 3: Adjusted for the same variables as model 2 and * diabetes (yes, non, unknown), percentage of calories from the other major nutrient groups, ** percentage of calories from the other major nutrient groups, # percentage of calories from polysaccharides, $ percentage of calories from monosaccharides, ¥ percentage of calories from vegetable protein, £ percentage of calories from animal protein, § percentage of calories from monounsaturated and polyunsaturated fats, ¶ percentage of calories from saturated and polyunsaturated fats ß percentage of calories from saturated and monounsaturated fats.

**Table 2 ijerph-18-00084-t002:** Effect of replacing 100 kcal of a nutrient with 100 kcal of another nutrient.

	Substituted Nutrient		
	Carbohydrates	Proteins	Fats
	HR 95% CI	HR 95% CI	HR 95% CI
Carbohydrates		1.09 (0.78 to 1.52)	1.02 (0.90 to 1.16)
Proteins	0.94 (0.67 to 1.33)		0.96 (0.68 to 1.35)
Fats	0.99 (0.87 to 1.12)	1.07 (0.77 to 1.49)	

**Table 3 ijerph-18-00084-t003:** Effect of replacing 100 kcal of a nutrient with 100 kcal of another nutrient stratified by menopausal status, BMI and ER status.

	Substituted Nutrient					
	Premenopausal Women			Menopausal Women		
	Carbohydrates	Proteins	Fats	Carbohydrates	Proteins	Fats
	HR 95% CI	HR 95% CI	HR 95% CI	HR 95% CI	HR 95% CI	HR 95% CI
Carbohydrates		1.34 (0.68 to 2.71)	0.83 (0.63 to 1.10)		0.84 (0.56 to 1.27)	1.03 (0.89 to 1.19)
Proteins	0.63 (0.30 to 1.32)		0.58 (0.27 to 1.24)	1.16 (0.77 to 1.73)		1.19 (0.80 to 1.76)
Fats	1.21 (0.90 to 1.63)	1.61 (0.77 to 3.37)		0.97 (0.84 to 1.13)	0.83 (0.56 to 1.25)	
	BMI ≤ 25			BMI > 25		
Carbohydrates		1.11 (0.60 to 2.08)	1.30 (1.01 to 1.68)		0.94 (0.60 to 1.49)	0.94 (0.80 to 1.10)
Proteins	0.85 (0.45 to 1.63)		1.12 (0.59 to 2.16)	1.11 (0.70 to 1.76)		1.04 (0.66 to 1.63)
Fats	0.76 (0.59 to 0.98)	0.85 (0.45 to 1.60)		1.07 (0.91 to 1.25)	1.00 (0.65 to 1.56)	
	ER negative			ER positive		
Carbohydrates		1.87 (0.80 to 4.38)	0.96 (0.76 to 1.22)		0.92 (0.63 to 1.36)	0.97 (0.83 to 1.30)
Proteins	0.56 (0.24 to 1.34)		0.54 (0.23 to 1.24)	1.12(0.75 to 1.66)		1.08 (0.72 to 1.61)
Fats	1.04 (0.83 to 1.31)	1.92 (0.86 to 4.30)		1.04 (0.88 to 1.21)	0.96 (0.65 to 1.42)	

Adjusted for: hospital of recruitment, age, PPS score (0, IA, IB, IIA, IIB, IIA, IIB, IIC, IV, missing), systemic treatment received by the patients: chemotherapy (yes, non), hormone therapy (yes, non) and immunology therapy (yes, non), total energy intake (Kcal/day) one year before the diagnosis, menopausal status, socioeconomic status (low, middle, high), education attained, physical activity (metabolic equivalents (METs)) during the 5 years before diagnosis, smoking status one year before recruitment (never; former; current) and Body Mass Index (kg/m^2^).

## Data Availability

The database was registered with the Spanish Agency for Data Protection, number 2102672171. Permission to use the study database will be granted to researchers outside the study group after revision and approval of each re-quest by the Steering Committee.

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
