# Peer review of "Dietary Constituents: Relationship with Breast Cancer Prognostic (MCC-SPAIN Follow-Up)"

_ijerph, 2020, doi:10.3390/ijerph18010084_

Round 1
Reviewer 1 Report
The work of Dierssen-Sotos et al. is extremely relevant for women due to the identification of the relationship between dietary constituents and BC prognosis, being modulated by hormonal status. However, there are a few things that I considered important noting:
- The introduction section could be more explored. The effects of diet on cancer are bigger than IGF-1 signaling. The article demonstrates that fat consumption increases survival. I think it is important to describe the effects of the different types of fat in BC. Additionally, also describe the types of fiber that modulate BC.
- In line 75, it is written "monosaturated" - please, correct.
- This paper investigated macronutrients regarding their main characteristics, but today, we know that the same type of nutrient can display different effects (for instance, omega-6 and omega-3 display opposite effects). I think it is important to analyze the BC outcomes in relation to these more refined characteristics.
- Regarding the tables displayed throughout the manuscript, I think it is better to visualize them in portrait.
- The FFQ should appear as supplementary material.
Author Response
Please, see the attachment file.

Reviewer 2 Report
Comments are attached

Reviewer 3 Report
Thank you very much for the opportunity to review the manuscript titled “Dietary Constituents: Its Relationship with Breast Cancer Prognostic (MCC-SPAIN Follow-Up)”, submitted by Trinidad Dierssen-Sotos and colleagues for consideration in Int. J. Environ. Res. Public Health.
This study was performed among 1350 women with all-stages breast cancer from 18 hospitals from the MCC-Spain breast cancer follow-up study, a prospective cohort study that is part of the MCC-Spain project. Participants completed a 140-item Food Frequency Questionnaire pre-diagnosis of breast cancer.
During a median follow-up of 6.5 years, 171 deaths were observed. None of the nutrients analysed was associated with mortality in the whole sample. However, some impact of specific nutrients was observed in subgroup analyses.
I have a number of minor and major remarks about this manuscript:
ABSTRACT
- Specify this was early breast cancer setting vs advanced
- Specify which cohort you are using
- Specify which type of Food Frequency Questionnaire you are using
- Specify what you mean by tertiles of intake of monosaccharides, what is the reference category, what tertile the HR that you present refer to
- Clarify what the HR and p value for polyunsaturated fats refer to
- Line 53: “these women” should read as “normal-weight women” (also clarify what is “normal”, I assume WHO normal BMI 18.5-25.5 Kg/m2 (?))
- Line 53, why do you state “On the other hand”, when the HR is <1 and you show a protective effect of high intake of fats on outcome, similarly to the HR that you present among all normal-weight women?
- Please clarify how “high intake” was defined in your study
- The sentence “According to our results, none of the different types of nutrients evaluated seems to have a direct effect on BC survival.” (line 56) Is redundant with “None of the nutrients analysed was associated with mortality in the whole sample” (line 47)
- Overall, abbreviations should be spelled out at first use
METHODS
- Was this an overall survival analysis or breast cancer specific analysis?
- Women with all subtypes and stages of breast cancer were included. These are important prognostic factors. Were Cox models for survival adjusted for these factors? Authors should give adequate explanation on the rationale they used to choose adjustment factors. They performed three iterations of models subsequently adjusting for additional variables, but the logic behind this is not completely clear to me. Please provide an adequate explanation. In addition there is a high number of missing in this cohort relative to stage (n=422) + non-applicable (because of neoadjuvant therapy).
- Also provide more details about why you decided to stratify analyses by menopausal status, BMI, and ER status. Were interaction terms studied between these factors and nutrients on OS in the whole cohort before stratification? Authors should also provide more background data and rationale (they state that the topic was not well studied in published literature but more details should be provided to contextualize the readers)
- Please also add more details on the substitution analysis that you performed. Is this a theoretical model?
- I have concerns that this analysis is biased by pooling all breast cancer stages together. 52/716 recurrence events were, as expected among stage I-II breast cancers. There were 2% of stage IV breast cancers, authors report “Non applicable (Neoadjuvancy)” cases among 121(8.96%) of patients. These data should be better explained. I am also not sure why authors did not decide to restrict the analysis among patients with early-stage breast cancer.
- Can authors explain why they show events as deaths/woman-years in tables?
SUPPLEMENTARY TABLE 1
- Can you please explain the inconsistency between n=67 HER2 intrinsic subtype (first row) vs. n=226 positive HER2 (last row)
DISCUSSION
I think this study should be discussed in the context of current data on the topic.
Two randomized trials addressed the impact of dietary modification on BC recurrence and mortality in women with early-stage BC, with mixed results: the Women Interventional Nutrition Study (WINS) and the Women’s Healthy Living and Eating Study (WHEL), both randomizing patients to a low-fat dietary intervention versus control. Although both studies were successful in affecting dietary change in study participants, WINS demonstrated an impact of the dietary intervention that led to a 24% reduction in the risk of BC recurrence(Chlebowski et al. 2006), while results were not confirmed in the WHEL study (Pierce et al. 2007). More recently, the Women’s Health Initiative Low Fat Dietary Modification (WHI DM) Trial(Chlebowski et al. 2017) randomized women to a dietary intervention designed to reduce fat intake and increase daily servings of fruits and vegetables, or to a usual diet control group. The trial showed a lower risk of all-cause mortality and a trend toward a lower rate of BC-specific mortality in women who developed BC while participating in the low-fat dietary intervention versus the control group. In addition, observational data also consistently linked poor dietary quality to increased risk of developing and dying from malignancy (Arnold et al. 2015; Calle et al. 2003; Van Dam et al. 2008; Lee et al. 2012; MacMahon et al. 2009)
Overall remarks:
- Originality/Novelty:The question is not extremely original and was already investigated in observational studies and RCTs
- Significance:Results should be reviewed in light of the remarks and concerns that I expressed in the relative section including comments on methodology
- Quality of Presentation: Improvements in consistency and clarity can be made
- Scientific Soundness: please see comments on methods
- Interest to the Readers:I think the study is appropriate to the readership of the journal
- Overall Merit: The authors should address major methodological limitations that undermine the credibility of the present version of the manuscript
- English Level: Although the manuscript reads well in the English language and I do not recommend that it undergoes review by a formal language editing service, I would suggest careful review of syntax and correction of minor language issues.
Round 2
Reviewer 1 Report
The authors answered all queries and improved significantly the quality of the manuscript.Author Response
Please see the attachment.

Reviewer 2 Report
The author group has addressed the major concerns with substantial inputs and at this present, the manuscript is a much more appropriate version.
Reviewer 3 Report
The authors addressed most of the comments that I had provided during the previous revision.
Some additional remarks below:
- Please consider using “neoadjuvant” rather than “neoadiuvancy”
- Line 219: “We have carried out a sensitivity analysis excluding advanced BC (stage IV) at diagnosis and no relevant differences were found therefore are not displayed”. The sentence “no relevant differences were found therefore are not displayed” belongs to results and should be reported in the relevant section
- Please see below the references of the cited studies:
- Arnold, Melina et al. 2015. “Global Burden of Cancer Attributable to High Body-Mass Index in 2012: A Population-Based Study.” The Lancet Oncology 16(1): 36–46.
- Calle, Eugenia E., Carmen Rodriguez, Kimberly Walker-Thurmond, and Michael J. Thun. 2003. “Overweight, Obesity, and Mortality from Cancer in a Prospectively Studied Cohort of U.S. Adults.” New England Journal of Medicine 348(17): 1625–38.
- Van Dam, Rob M. et al. 2008. “Combined Impact of Lifestyle Factors on Mortality: Prospective Cohort Study in US Women.” BMJ 337(7672): 742–45.
- Lee, I. Min et al. 2012. “Effect of Physical Inactivity on Major Non-Communicable Diseases Worldwide: An Analysis of Burden of Disease and Life Expectancy.” The Lancet 380(9838): 219–29.
- MacMahon, S. et al. 2009. “Body-Mass Index and Cause-Specific Mortality in 900 000 Adults: Collaborative Analyses of 57 Prospective Studies.” The Lancet 373(9669): 1083–96.
